# The Effects of Short-Term and Long-term Hearing Changes on Music Exposure: A Systematic Review and Meta-Analysis

**DOI:** 10.3390/ijerph17062091

**Published:** 2020-03-21

**Authors:** Sunghwa You, Tae Hoon Kong, Woojae Han

**Affiliations:** 1Laboratory of Hearing and Technology, Research Institute of Audiology and Speech Pathology, College of Natural Sciences, Hallym University, Chuncheon 24252, Korea; shyouuu@gmail.com; 2Division of Speech Pathology and Audiology, College of Natural Sciences, Hallym University, Chuncheon 24252, Korea; 3Department of Otorhinolaryngology-Head and Neck Surgery, Yonsei University Wonju College of Medicine, Wonju 26426, Korea; cochlear84@gmail.com

**Keywords:** music-induced hearing loss, personal listening device, music exposure, young adults, pure tone thresholds, otoacoustic emissions

## Abstract

The present study explores the scientific evidence on whether music exposure temporarily or permanently affects hearing sensitivity in young adults. Six electronic databases were searched using related keywords for the four categories of personal listening devices, listening habits, hearing outcomes, and age. The Hedges’ g and its 95% confidence intervals (CIs) were estimated. A Higgins *I*^2^ was also used to check for heterogeneity. To test for publication bias, funnel plots were drawn using Egger’s regression. Based on the inclusion criteria, 16 studies were divided into two groups to identify short-term hearing changes (*n* = 7) and long-term hearing changes (*n* = 9). In the short term, there was no significant immediate change in the thresholds or amplitudes after the music exposure, although pure-tone thresholds (PTAs) and distortion product otoacoustic emissions (DPOAEs) did show the highest effect size (−0.344, CI −0.727 to 0.038) and (0.124, CI −0.047 to 0.296) at 4 kHz. On the other hand, for long-term hearing changes, the PTA provided the highest effect size at 6 kHz (−0.525, CI −0.897 to −0.154) and 8 kHz (–0.486, CI −0.819 to −0.152), while also implying that habitual and repeated personal listening device (PLD) usage can act on some significant hearing changes in audiological tests. We conclude that the use of a PLD produces a few temporary hearing changes at 4 kHz after its use but that the changes are then reversed. However, it is important to note heavy PLD users’ experience regarding permanent changes in their hearing thresholds at high frequencies, and the public should be educated on this issue.

## 1. Introduction

Excessive noise exposure, which usually causes damage to the ear’s hair cells, is considered a major cause of hearing disorders worldwide [1]. According to current scientific reporting, nonoccupational noise exposure, such as recreational noise, occurs in as much as 21% of hearing disorders, as opposed to occupational noise at less than 16% [1]. Thus, many contemporary researchers have become concerned about the hearing loss induced by listening to music on personal listening devices (PLDs) in public [2,3].

PLDs have become increasingly popular over the last two decades, and their form has changed from a portable CD player, MP3 player, or iPod [4] to a mobile phone (or smartphone) with a free MP3 player function [5]. With their rapid development, PLDs allow users to listen to high-quality music for an unlimited time at a high volume while providing more listening convenience; that is, they are easy to carry, have longer battery life, and offer larger music storage capacity [6].

Notably, potentially substantial evidence in the literature shows an increasing potential risk from excessive music exposure to young adults’ listening to their music using PLDs [3,5]. In addition, the prevalence of hearing loss with music exposure in young adults ranges from 9.3% to 14%, depending on the study or nation [7]. In addition, several studies have demonstrated that maximum outputs of PLDs can exceed 125 dBA [8]. The preferred listening levels (PLLs) adopted by young users range from 71 to 105 dBA [5,9,10], but approximately 58% of adolescents and college students have exceeded that recommended maximum noise exposure [11], and the prevalence of increased hearing thresholds due to public music exposure via PLDs has not been overlooked [12]. Furthermore, various studies have reported an association between exposure to music at a high intensity and hearing deterioration, including temporary threshold shifts (TTS), tinnitus, hyperacusis, and recruitment, all of which eventually produce permanent hearing loss to the users [2,13,14]. In other words, such hearing deterioration seems to be temporary and quickly recovered at an early stage of music exposure; however, these temporary changes can naturally develop into a permanent hearing loss from habituated and accumulated exposure [12] and then accelerate into an age-related hearing loss [15].

In summary, it is worth exploring the existing evidence to compare between temporary (short-term) and permanent (long-term) hearing changes; by doing so, we can better understand the different aspects of young adults’ hearing changes resulting from excessive music exposure to achieve better public health. Moreover, it is important to characterize the early identification of excessive music exposure on hearing, if any, using a combination of tools that can detect hearing damage at the initial subclinical stage. In this light, the purpose of the present review and meta-analysis is to review new and timely literature for a comparison of the short-term and long-term hearing changes in young PLD users who use these devices often and heavily.

## 2. Materials and Methods

### 2.1. Strategy for Systematic Search

To provide transparency in the systematic search and selection, the present study followed the methodology and protocols of the preferred reporting items for the Systematic reviews and meta-analyses (PRISMA) statement [16] and International prospective register of systematic reviews (PROSPERO), part of the National Institute for Health Research (NIHR) [17]. Consequently, the study pre-established data extraction and analysis using inclusion and exclusion criteria for its systematic review procedure (see Figure 1).

To identify articles for inclusion, six electronic databases (i.e., Embase, Medline, PubMed Central, Web of Science, Scopus, and the cumulative index to nursing and allied health (CINAHL)) were searched in October 2019 using the key terms listed in Appendix A. The terms were divided into four groups: (1) PLDs, (2) listening intensity and duration through PLDs, (3) related hearing conditions and symptoms, and (4) age. The terms associated with each group were then listed by the Boolean term “OR” and the resulting four groups were combined using the Boolean term “AND.” The final search was conducted on 25 October, 2019.

### 2.2. Selection Criteria

All inclusion criteria were consistent in terms of the participants, intervention, control, outcomes, and study design (PICOS) strategy [18]. The PICOS criteria are described in Table 1. Articles that did not correspond to the topic that were not journal articles (letter, book, conference proceeding, only including the abstract and pilot study), or were not peer-reviewed studies and/or written in English were excluded.

### 2.3. Selection Process for Studies and Data Extraction

Based on combinations of the key terms in different databases (see Appendix A), the articles with potential relevance were identified by carefully investigating the titles and abstracts, while removing duplicate articles. The two authors independently screened all searched articles for eligibility in terms of specific inclusion and exclusion criteria. In the case of inconsistency in interpretation, the authors discussed the issue until the criteria were consistent and then selected the final analysis subjects. Subsequently, the authors assessed the full text of selected articles with a thorough compliance verification following the PICOS criteria. The review and systematic extraction of the data from the studies were included in the systematic review.

### 2.4. Assessment of the Quality of the Studies

To assess the quality of the selected studies, the Newcastle–Ottawa scale (NOS) for cohort study [19] was used in the present study. The NOS consists of three subcategories (i.e., selection, comparability, and outcomes) with eight items in total. A star system was used to allow a quantitative assessment of study quality, with one star for each item, with the exception of item-related comparability that allowed the assignment of two stars. As a result, the NOS quantitative measure ranged between zero and nine stars.

### 2.5. Statistical Analysis

In the present study, a statistical software program for meta-analysis, Comprehensive meta-analysis, (Version 3; Biostat, Englewood, NY, USA) was used. After the selected studies were divided according to the designated study design, a meta-analysis was carried out to compare changes in the effect size for both hearing thresholds of pure-tone audiometry (PTA) and amplitude of otoacoustic emissions (OAEs) for the experimental and control groups, for both short-term and long-term hearing changes. For each meta-analysis, the effect size was calculated as Hedges’ g, along with the 95% confidence interval (CI) with a *p* < 0.05 significance set. The heterogeneity between the studies was calculated as the *I^2^*. The degree of heterogeneity was set at low (*I^2^* <40%), medium (40% < *I^2^* <60%), and high (60% < *I^2^*) levels. A randomized model was applied to estimate the combined effect. To determine the source of any heterogeneity, a meta-regression analysis was also performed. For additional insight, subgroup analyses were performed based on the PLLs and exposure times. Any small study effects were explored using funnel plots and Egger’s regression.

## 3. Results

### 3.1. Selection Outcomes

Using the search strategy, 2505 records were identified in the six electronic databases. After removing 477 duplicates, 1859 articles were removed based on the exclusion criteria, such as not being peer-reviewed, not an academic article, or irrelevant topic. A total of 169 remaining articles that were potentially relevant were screened by carefully reading their titles and abstracts. Using this process, 152 articles were excluded due to noncompliance with PICOS criteria and 17 articles were included. Figure 1 illustrates these processes.

In Table 2, the Newcastle–Ottawa scale (NOS) quality evaluation is provided for all 17 studies. The authors confirmed that four [9,20,21,22] of the selected studies had “good” quality (scored 7–9). The remaining studies were evaluated as having “fair” quality (score 4–6) [20,21,22,23,24,25,26,27,28,29,30,31,32,33,34] except for one study with a score of 3, evaluated as having “poor” quality [35]. Thus, 16 articles were included for systematic review and meta-analysis in this study.

### 3.2. Study Characteristics

Sixteen studies were included in the systematic review and meta-analysis. These studies were cohort studies or cross-sectional studies and investigated the hearing thresholds in young adults. More specifically, the 16 studies were divided into two groups: temporary (or short-term; within 24 h) hearing changes and permanent (or long-term) hearing changes.

The studies with short-term hearing changes (*n* = 7) assessed the change in thresholds of PTA or amplitude of distortion product otoacoustic emissions (DPOAEs) of pre- and post-music exposure. The studies related to long-term hearing changes (*n* = 9) compared the PTA and DPOAE test outcomes of the PLD user group and nonuser group, as divided by questionnaires or interviews. Additionally, both exposure intensity and time (or duration) of PLD use were checked. Each study is summarized in Table 3 and Table 4 for short-term and long-term hearing changes, respectively.

### 3.3. Meta-Analysis of Music Exposure Effects with Subgroup Analysis

Sixteen high-quality studies that reported hearing thresholds of PTA (4, 6, and 8 kHz testing frequencies) and amplitudes of the DPOAEs (2, 4, and 6 center frequencies) were pooled for analysis. As mentioned earlier, the studies were divided into two groups to see short-term hearing changes (pre/post-exposure with 24 hours; *n* = 7) [20,23,25,27,28,29,32] and long-term hearing changes (PLDs users/non-users; *n* = 9) [9,21,22,24,26,30,31,33,34].

For short-term hearing changes within 24 h after music exposure [20,23,25,27,28,29,32], the studies pooled for analysis showed PTA 4, 6, and 8 kHz changes at pre- and post-music exposure (see Figure 2). The thresholds for PTA at 4 kHz showed the highest effect size (−0.344, 95% CI −0.727 to 0.038, *p* 0.078), while the distribution of the 4 kHz thresholds showed the highest heterogeneity (*I^2^* 74.90%). Interestingly, the thresholds at 6 and 8 kHz showed different aspects of effect size. The effect sizes of 6 and 8 kHz thresholds appeared relatively smaller; 6 kHz, −0.059, CI −0.438 to 0.320; *p* 0.760 and 8 kHz, −0.066, CI −0.436 to 0.303; *p* 0.725. Moreover, the 6 and 8 kHz thresholds showed low heterogeneity (less than 5% of *I^2^*). Regardless of the test frequencies, however, the overall effect size of the PTA thresholds showed –0.154 (CI −0.393 to 0.084; *p* 0.205) with medium heterogeneity (*I^2^* 42.92%). These results did not indicate any significant deterioration of the hearing thresholds after music exposure.

Similar to PTA outputs, the amplitude of DPOAEs showed the largest effect size also at 4 kHz (0.124, CI −0.047 to 0.296; *p* 0.155) (see Figure 3). The heterogeneity showed 0% of *I^2^*, regardless of the test frequencies. Although amplitude at 6 kHz showed a similar effect size with 4 kHz (0.106, CI -0.065 to 0.278; *p* 0.224), the amplitude at 2 kHz was relatively less affected (0.029, CI −0.154 to 0.212; *p* 0.755). The overall effect size of the DPOAEs was 0.089 (CI −0.012 to 0.190; *p* 0.080, *I^2^* 0%), indicating no significant amplitude change within 24 h after music exposure. In sum, the short-term result after music exposure did not show any meaningful changes for PTA thresholds or DPOAE amplitudes.

To identify the long-term hearing changes after music exposure [9,21,22,24,26,30,31,33,34], the pooled studies for analysis are shown in Figure 4 and Figure 5 for PTA and DPOAE, respectively, in the same frequencies along with the short-term hearing changes.

In the PTA testing, the highest effect size was at 6 kHz (−0.525, CI −0.897 to −0.154) and significant (*p* 0.006), although the data for 6 kHz showed the highest heterogeneity (*I^2^* 80.09%). The 8 kHz also had a –0.486 significant effect size (CI −0.819 to −0.152; *p* 0.004) with medium heterogeneity (*I^2^* 57.62%). Unlike the short-term hearing changes, however, the thresholds for 4 kHz showed the lowest effect sizes (−0.266, CI −0.598 to 0.066; *p* 0.116) and heterogeneity (*I^2^* 38.63%). Overall, the long-term hearing changes determined by PTA measurement showed a −0.419 effect size (CI −0.632 to −0.207) with significance (*p* 0.000), indicating that consistent PLD usage significantly and negatively affects the user’s hearing thresholds in the long term.

With the DPOAEs, the highest effect size was at 2 kHz (0.135, CI −0.127 to 0.398; *p* 0.311) and 4 kHz (0.131, CI −0.131 to 0.393; *p* 0.329) with medium (*I^2^* 58.77%), and high (*I^2^* 76.65%) heterogeneity, respectively. However, the long-term effects of music exposure on DPOAEs showed an overall 0.115 effect size (CI 0.037 to 0.266) with no significance (*p* 0.138). In short, analysis of the long-term changes showed a significant negative effect for the user’s hearing thresholds, especially at 6 and 8 kHz of PTA, but no remarkable effect to the amplitude of DPOAEs.

In addition, the four funnel plots that related to the results of PTA thresholds and DPOAEs amplitude showed no publication bias (see Figure 6) with no significant Egger’s regression (*p* > 0.05).

## 4. Discussion

The present review focused on: (1) exploring the existing evidence for a comparison between the short-term and long-term hearing change that occurs from the music exposure, and (2) characterizing these aspects of changed hearing status for early identification and intervention. Thus, the present study examined PLD users’ hearing status via the most sensitive clinical tests, such as PTA and DPOAE, and elucidated a possible association with the exposure to music. 

Similar to the present study, the World Health Organization (WHO) reported guidelines for leisure noise with regulation of the exposure intensity and time, health outcomes, and interventions [36]. In its assessment of the evidence on hearing loss, the WHO determined that: (1) the effect of regular music exposure cannot be quantified, (2) the effect of the specific threshold analysis is limited, and (3) analysis of the pooled effect size is limited due to lack of data from previous studies. Although the present study cannot resolve all the limitations in the guidelines, it can present insight into behavioral and objective hearing outcomes at a specific frequency based on a meta-analysis.

### 4.1. Short-term Hearing Changes Due to Music Exposure

Seven articles indicating pre- and post-experimental study were categorized as short-term hearing changes [20,23,25,27,28,29,32]; thresholds and amplitude data from 300 young adults (ages 17 to 38; 20 to 132 participants per study) were measured by PTA [20,22,24,27,29] and DPOAE [20,23,25,27,28,29] before and after their music exposure through PLDs with earphones, while also analyzing their exposure intensity and exposure time.

In terms of PTA thresholds, three of the five studies reported no statistically significant change in hearing thresholds after the music exposure [23,25,29], whereas two reported a small but reliable, temporary threshold shift [24]. These users had a significantly reduced hearing threshold after music exposure [25]. However, a significant variation in audiological results, exposure intensity, and exposure time among the studies also existed. With some variation, four of the seven studies reported that the change was very limited and not significant in the DPOAE amplitude after the music exposure [25,27,28,29], whereas the other three reported significant changes in the same DPOAEs amplitude [20,23,32].

When focusing on exposure intensity and time, in four studies [20,23,25,32] participants were exposed to a high intensity of 85 dB (C) to 100.3 dB(A) with a long exposure duration of 30 min to 4 h. However, the measured units were different, and this limited generalization. On the other hand, the studies that reported no significant changes were exposed to music levels from 50.8 to 98.7 dBA for 30 min to 1 h, which perhaps was not a sufficiently strong music level or a long enough time to cause significant threshold changes in healthy young adults [27,28]. Although limited to a direct comparison to WHO’s regulation [36] due to heterogeneity, such as the difference in units and time, only two studies were comparable [25,27]. Based on *L*_Aeq,1h_, Keppler, and Bockstael reported 92.41 dB *L*_Aeq,1h_ at 1 h [25], and Keppler et al. reported 86% of subjects exposed at greater than 92.25 dB *L*_Aeq,1h_ at 1 h [27]. Considering the hazardous exposure intensity and time in the regulation (95 dB *L*_Aeq_, 1 h) [7,10], these results indicated no significant change in PTA or DPOAE and were consistent with the regulations.

### 4.2. Long-term Hearing Changes Due to Music Exposure

In the analysis of long-term hearing changes, nine studies using either a case-control study or a cross-sectional study were grouped [9,21,22,24,26,30,31,33,34]. The participants in these studies were assigned according to their PLD user group (or high-risk group) and non-PLD group based on the results of the questionnaires, interviews, and participants’ usual listening levels. Then, their PTA thresholds [9,21,26,30,31,33,34] (three studies were also tested for an extended testing frequency up to 16 kHz) [9,26,31], and DPOAE amplitudes [9,21,25,35] were compared for music exposure intensity and use period for the PLDs. The variation in the long-term hearing changes was smaller than the short-term effects. 

Based on the PTA thresholds [9,24,26,30,31,33,34], four of the seven studies reported that the hearing thresholds of PLD users were significantly higher than those of non-PLD users at the conventional testing frequencies of 0.25 to 8 kHz [26,30,33,34]. Of the other three studies, two reported that PLD users had significantly poorer hearing thresholds at the extended testing frequencies of 9 to 16 kHz compared to the thresholds for the non-users [9,31]. Thus, six studies concluded that significant changes in the hearing thresholds were caused by PLD usage [9,26,30,31,33,34]. Fortunately, four studies additionally investigated subjective hearing symptoms, such as tinnitus, difficulty with communication, and ear pain after music exposure. For example, Sulaiman et al. [9] reported that immediately after music exposure, 37.1% to 51.5% of the PLD users experienced, in order of appearance, ear pain, tinnitus, and difficulty hearing others. Moreover, these hearing symptoms became more frequent when the listening intensity was higher [26]. Widen et al. [34] also reported that 7% to 8% of their subjects experienced hearing symptoms like tinnitus, sound sensitivity, and sound fatigue. Torre and Reed [22] supported this finding in that 12.9% to 24.8% of their respondents also experienced tinnitus or difficulty in communication. 

Regardless, the current results of the DPOAE meta-analysis did not clearly support the studies mentioned above; two of the four studies reported no significant differences at DPOAE amplitude [23,24], while the other two showed a significantly decreased amplitude [9,21] although still having some variation. As one possibility to explain these contradictory findings, the studies have differences in intensity and time. For example, Kumar et al. [25] had a similar intensity to Sulaiman et al. [9] at 76 dBA *L*_Aeq,8h_, but they showed no significant differences. The other study that reported no significant difference showed relatively low intensity (73.0 dBA *L*_Aeq_) with a short use period (1 h/day) [22]. The studies that reported significant differences in DPOAEs showed relatively high intensity (81.3 dBA, 8h 76.2 dBA *L*_Aeq_ [9], and 95.5 dBA [21]) with a long use period (2.7 h/day [8] and 2.9 h/day [21]). We have an important message in that changes can be caused in a relatively short use period (1.5 h/day). Similar to the investigations of the short-term effect, only three studies were compared against the WHO regulation [36]. In terms of PTA, based on *L*_Aeq_, Kumar and Deepashree reported 87.97 dB *L*_Aeq_ at 2 h/day [31], and Hussain et al. reported 83 dB *L*_Aeq_ at 1.99 h/day [33]. This is in comparison to the regulation (74 dB *L*_Aeq_, 2 h/day). These results exceed the regulation and are consistent with the significant poor PTA results. Furthermore, Torre and Reed reported 73.0 dBA *L*_Aeq_ at 1 h/day, which did not exceed the regulation, with no significant DPOAE amplitude [22].

### 4.3. Limitation of the Study

The present study had limitations in terms of inconsistency in units. The intensity data of the pooled studies which we selected in this systematic review could not be synthesized because of the heterogeneity of outcome measurement (specifically the measurement unit), and the measurement methods varied depending on the study. More specifically, the intensity results in various studies (e.g., dB SPL, dBA, dBC, *L*_Aeq_) could not be easily integrated due to the different weights and temporal averaging according to frequency when first measured. However, it might be interesting to integrate the different measured intensity units for each study to use meta-analysis, so further investigation should be considered.

## 5. Conclusions

This review and analysis showed that PLD users experience a few temporary hearing changes at 4 kHz, but their hearing is restored. However, if these musical exposures are repetitive and cumulative, then these users will experience permanent hearing changes or loss, especially if exposure is at high frequencies; of further concern is that young users and/or the general public do not yet recognize the specific standards that can prevent nonoccupational noise-induced hearing loss. Therefore, it is desirable to campaign on and communicate to the public the hazards of music exposure and its negative effects on hearing in terms of public health when using PLDs, as the analysis in our study supports. In addition, health professionals have a clear opportunity to influence the public’s hearing behavior by providing appropriate education on hearing loss prevention regarding the ongoing recreational noise exposure to music via PLDs at all levels of society.

## Figures and Tables

**Figure 1 ijerph-17-02091-f001:**
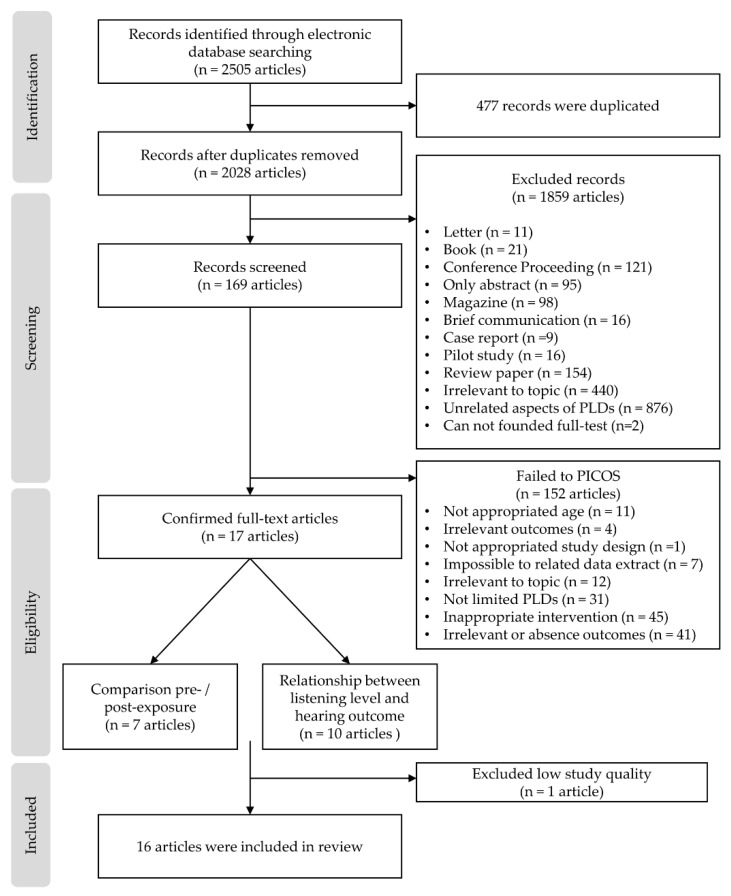
Article selection process using the inclusion and exclusion criteria for the present study. Notes: PICOS, participants, intervention, control, outcomes, and study design.PLD: personal listening device.

**Figure 2 ijerph-17-02091-f002:**
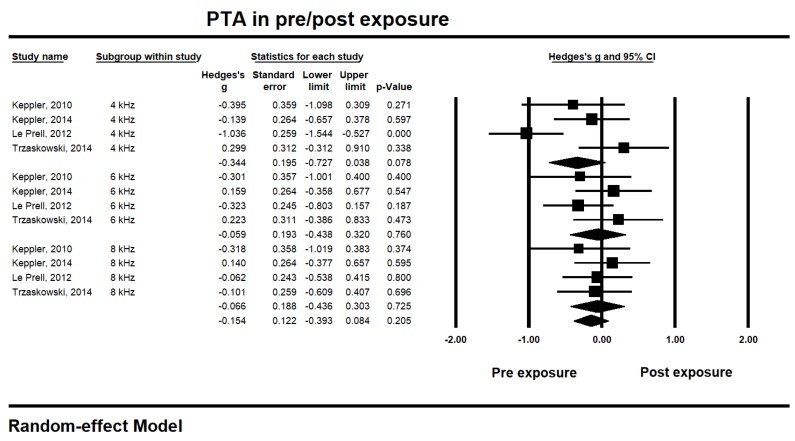
Short-term hearing changes for the pure tone audiometry (PTA) thresholds at 4, 6, and 8 kHz within 24-h after music exposure.

**Figure 3 ijerph-17-02091-f003:**
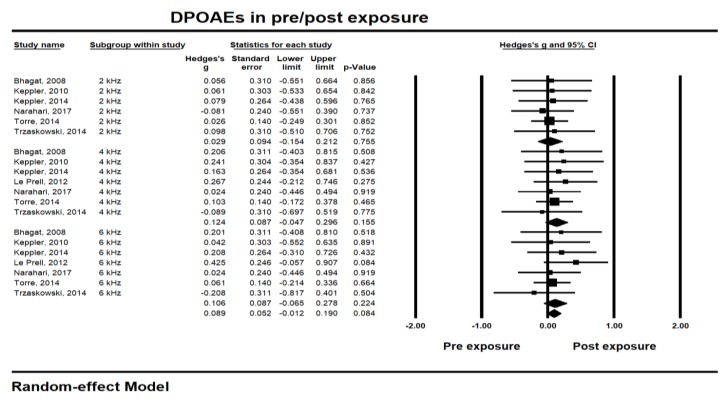
Short-term hearing changes in distortion product otoacoustic emissions (DPOAEs) amplitude at 2, 4, and 6 kHz center frequencies within 24 h after music exposure.

**Figure 4 ijerph-17-02091-f004:**
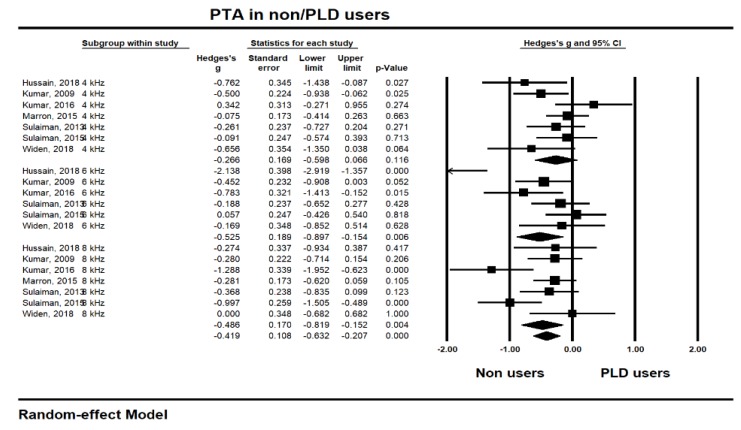
Long-term hearing changes of pure tone audiometry (PTA) thresholds at 4, 6, and 8 kHz after music exposure.

**Figure 5 ijerph-17-02091-f005:**
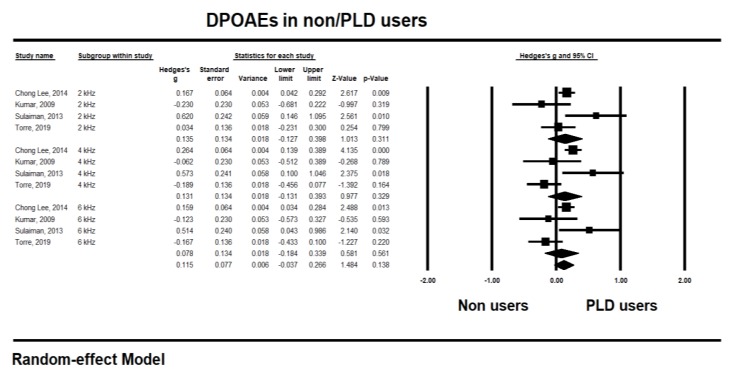
Long-term hearing changes for the distortion product otoacoustic emissions (DPOAEs) amplitude at 4, 6, and 8 kHz center frequencies after music exposure.

**Figure 6 ijerph-17-02091-f006:**
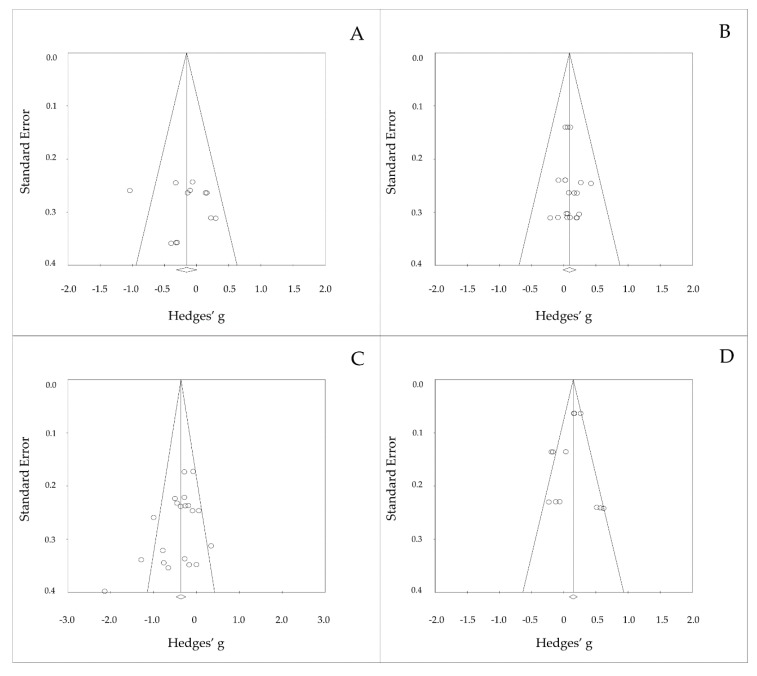
Funnel plotting of the results for short-term and long-term hearing changes to PTA or DPOAEs when using PLDs; (**A)** PTA in short-term effect, (**B**) DPOAEs in short-term effect, (**C)** PTA in long-term effect, and (**D**) DPOAEs in long-term effect.

**Table 1 ijerph-17-02091-t001:** Inclusion criteria based on PICOS strategy.

PICOS	Content Category
Participants	Young adults using PLDs
Intervention	Device exposure intensity and/or time
Control	Comparison to a control group or repeated measures (pre- and post-PLDs exposure comparison).
Outcomes	1+ audiometry outcomes (PTA, OAEs, and EHF) and/or hearing symptoms
Study Designs	Cohort studies, cross-sectional studies, and repeated measures

Notes: PICOS, participants, intervention, control, outcomes, and study design; PLD, personal listening device; PTA, pure-tone audiometry; OAE, otoacoustic emission; EHF, expanded high frequency.

**Table 2 ijerph-17-02091-t002:** Scientific study validity criteria based on the Newcastle–Ottawa scale used for cohort studies.

Authors	Quality Assessment per the Newcastle–Ottawa Scale			
	Selection	Comparability	Outcome	Total
Bhagat, 2008 [23]	2	1	3	Fair (6)
Kumar, 2009 [24]	3	1	2	Fair (6)
Keppler, 2010 [22]	3	1	2	Fair (6)
McNeill, 2010 [23]	1	1	1	Poor (3)
Le Prell, 2012 [20]	3	1	3	Good (7)
Lee, 2014 [21]	4	2	2	Good (8)
Sulaiman, 2013 [9]	4	1	2	Good (7)
Sulaiman, 2015 [26]	3	1	2	Fair (6)
Keppler, 2014 [27]	3	1	2	Fair (6)
Torre, 2014 [28]	3	1	2	Fair (6)
Trzaskowski, 2014 [29]	2	1	3	Fair (6)
Marron, 2015 [30]	3	1	2	Fair (6)
Kumar, 2016 [31]	3	1	2	Fair (6)
Narahari, 2017 [32]	3	1	2	Fair (6)
Hussain, 2018 [33]	3	1	2	Fair (6)
Widén, 2018 [34]	3	1	1	Fair (5)
Torre, 2019 [22]	3	2	2	Good (7)

Note: The overall quality of the study evaluation was arbitrarily defined as “poor” (score, 0–3), “fair” (4–6), or “good” (7–9).

**Table 3 ijerph-17-02091-t003:** Summary of studies showing short-term hearing changes and included in the meta-analysis.

Authors (Publication Date)	Subjects (Ages)	Method			Main Outcomes
		Study Design	ExposureIntensity	Exposure Time (min)	
Bhagat and Davis (2008) [23]	20 young adults (18–38 y)	Hearing statuses were measured pre, post (30 min), and recovery (48 h) from time of music exposure PTA OAEs (DPOAEs, SOAEs)	85 dBC ± 3 dB	30	Although the hearing thresholds did not change, the changes in DPOAEs and SSOAEs were early warning signs of the harmful effects of high levels of music exposure on cochlear functioning.
Keppler and Bockstael (2010) [25]	Noise exposure group 15 young adults (19–28 y), Control group 28 young adults (19–28 y)	Audiometric results of both groups were compared PTA OAEs (DPOAEs, TEOAEs)	*L*_Aeq__,1h_ 92.41 dBA *L*_Aeq__,8h_ 83.38 dBA	60	In the noise exposure group, the PTA and TEOAE amplitudes changed significantly between pre- and post-exposure. However, these patterns were not found for the DPOAE amplitudes.
Le Prell et al. (2012) [20]	33 young adults (18–27 y)	Hearing status was measured pre, post (15min, 75min, 135min, 195min), and recovery (following 1 day and 1 week) after music exposure PTA OAEs (DPOAEs)	94.5 dBA (*n* = 10) 99.6 dBA (*n* = 11) 100.3 dBA (*n* =12)	240	Changes in hearing thresholds showed a notched configuration, largest at 4 kHz. After music exposure, small, but reliable, temporary threshold shifts were found, but these threshold shifts quickly recovered during the first 3 h after music exposure.
Keppler et al., (2014) [27]	28 young adults (19–30 y)	Hearing status was measured pre- and post- (immediately, 30 min) exposure PTA OAEs (DPOAEs, TEOAEs)	Based on *L*_Aeq,1h_ 82.52 dBA (*n* = 2) 87.46 dBA (*n* = 2) 92.25 dBA (*n* =12) 98.70 dBA (*n* = 12)	60	The authors reported that no clear relationship exists between temporary hearing deterioration and the amount of efferent suppression.
Torre and Grace (2014) [28]	Noise exposure group 101 young adults (18–30 y), Control group 21 young adults	Audiometric results for both groups were compared for pre- and post- (immediate) exposure DPOAE absolute levels and generator and characteristic frequency component levels were noted	50.8 dBA (*n* = 7) 56.6 dBA (*n* = 56) 58.8 dBA (*n* = 14) 62.3 dBA (*n* = 24)	60	Based on the lower listening levels, DPOAEs showed very little or indeed no significant change after music exposure.
Trzaskowski et al. (2014) [29]	20 young adults (22–27 y)	Audiological tests were conducted pre-, post (immediately), and recovery (24 h) after music exposure PTA OAEs (DPOAEs, TEOAEs)	86.6 dBA	30	No statistically significant changes were found in OAE and PTA following 30min of music exposure.
Narahari et al. (2017) [32]	34 young adults (17–21 y)	DPOAEs at pre- and post- (2 h) music exposure were analyzed	98.29 dBSPL	120	The DPOAEs were affected immediately post music exposure, especially at 8 kHz or higher.

Note: y, years old; PTA, pure tone audiometry; OAEs, otoacoustic emissions; DPOAEs, distortion product otoacoustic emissions; SOAEs, spontaneous otoacoustic emissions; TEOAEs, transient-evoked otoacoustic emissions.

**Table 4 ijerph-17-02091-t004:** Summary of studies showing long-term hearing changes included in the meta-analyses.

Authors (Publication Date)	Subjects (Ages)	Method				Main Results
		Study Design	Exposure Intensity	UseTime	UsePeriod (yr)	
Kumar el al. (2009) [24]	PLD user group 70 young adults (17–24 y) Non-PLD group (control) 30 young adults	After dividing into two groups based on the questionnaire, the audiological results of both groups were compared. PLL in quiet and noisy conditions PTA OAEs (DPOAEs)	76 dBA *L*_Aeq,8h_	1.5 h/day	>2 yr	No significant differences were found at PTA and DPOAE for either group. In terms of a correlation analysis, a positive correlation exists between PTA and PLLs, and a negative correlation exists between DPOAEs and PLLs.
Sulaiman et al. (2013) [9]	PLD user group 35 young adults (18–30 y) Non-PLD group (control) 35 young adults	After dividing into two groups based on the interviews, hearing status was compared. Interviews (hearing symptoms) PLLs in blinded volume setting PTA, EHFs OAEs (DPOAEs, TEOAEs)	81.3 dBA (76.2 dBA *L*_Aeq,8h_)	2.7 h/day	3.2 yr	In EHFs, the mean hearing thresholds of PLD users were significantly higher, while conventional PTA users did not have typical characteristics of NIHL. OAE (DPOAE, TEOAE) amplitudes in PLD users were reduced compared to those in the non-PLD group.
Lee et al. (2014) [21]	High-risk group DPOAEs (*n* = 279) TEOAEs (*n* = 325) Others group (ear) D POAEs (*n* = 2097) TEOAEs (*n* = 2462)	A listening habit survey was conducted, and the subjects were assigned to each group. Based on the group, the OAE results were compared between the two groups. Listening habit survey PLLs OAEs (DPOAEs, TEOAEs)	95.5 dBA	2.9 h/day	5.5 yr	• OAE (DPOAE, TEOAE) levels in high-risk group were significantly depressed. Specifically, TEOAE levels were lower at 4 kHz, and DPOAE levels decreased at 1, 2, 3, and 4 kHz.
Marron et al. (2015) [30]	180 young adults (17–25 y)	Based on three questionnaires, PLL and PTA results of the subjects were analyzed. 3 questionnaires PTA PLLs	73 dBA	9.51 h/week	10–12 yr	Listeners who use their PLDs for more than 7.5 h a week exhibited a statistically significantly worse hearing threshold.
Sulaiman e al. (2015) [26]	PLD user group 228 young adults (18–30 y) Non-PLD group (the control) 54 young adults	Audiological results were compared by group, PLL, and hearing symptoms. Interview (habits, symptoms) PTA, EHFs PLLs	70.1 dBA (62.4 dBA *L*_Aeq,8h_)	2.2 h/day	3.15 yr	In the PLD user group, 20.1% had >75 dBA (*L*_Aeq,8h_), while 4.4% had >85 dBA, which can cause hearing damage. The subjects who listened to their devices at more than 75 dBA showed a significantly higher hearing threshold and incidence of hearing symptoms.
Kumar and Deepashree (2016) [31]	PLD user group 30 young adults (15–30 y) Non-PLD group 30 young adults	When comparing the two groups, the effects of PLD use on hearing were evaluated. Questionnaire for group assignment PLLs PTA, EHFs OAEs (TEOAEs)	87.97 dBA *L*_Aeq_	2 h/day	> 2 yr	Subjects who listened to PLDs at higher than 80 dBA (*L*_Aeq_) showed poorer EHF thresholds and decreased TEOAE amplitude.
Hussain et al. (2018) [33]	50 young adults (19–39 y) High-risk group (subgroup) 11 young adults who listened to their devices at over 91 dBA	Based on the questionnaire, hearing status and listening preferences were compared. Questionnaire PLLs in quiet and noisy condition PTA	85.5 dBA (All subjects), 96.2 dBA (High-risk subgroup)	2 h/day	6.1 yr	PLL and hearing thresholds were significantly correlated. In addition, the high-risk subgroup of subjects revealed poorer hearing thresholds at 4khz and 6 kHz.
Widén et al. (2018) [34]	279 young adults (17 y)	Based on the questionnaire, output levels and hearing statuses were compared.QuestionnairePLLsPTA	83 dBA *L*_Aeq_	1.99 h/day	3.7 yr	• Subjects who listened at more than 85 dBA showed higher hearing thresholds.
Torre and Reed (2019) [22]	216 young adults (college-aged, mean age, 20.95)	As measured for questionnaire-related PLDs and objective and subjective outcomes, the effects of DPOAEs were analyzed. Questionnaire (listening habits) OAEs (DPOAEs)	73.0 dBA *L*_Aeq_	1 h/day	-	• In a subjective volume setting, 35% of subjects reported loud or very loud, but the measured PLLs were not considered hazardous. •There was no association between PLLs and DPOAEs, regardless of gender.

Note: y, years old; PLD, personal listening device; OAEs, otoacoustic emissions; NIHL, noise-induced hearing loss; DPOAEs, distortion product otoacoustic emissions; TEOAEs, transient-evoked otoacoustic emissions.

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
