# Peer review of "The Effects of Short-Term and Long-term Hearing Changes on Music Exposure: A Systematic Review and Meta-Analysis"

_ijerph, 2020, doi:10.3390/ijerph17062091_

Round 1

Reviewer 1 Report

This article deals with effects of short-term and long-term hearing changes on music exposure for young adults. Using strategy for systematic search, authors selected 16 articles and analyzed the context of these articles. They concluded that personal listening devices (PLDs) users seem to have a few hearing changes at 4kHz after use, and repetitive and cumulative use will connect to permanent hearing changes or loss. These main research findings will be important for understanding effects of music exposure in people. The study design, procedure, data analysis, and conclusions appropriate. Minor revisions may be needed.

Minor revision suggestions:

Some figures (from figure 2 to figure 5) is unclear and hard to understand. Could you organize the data in figures and provide additional explanation for easy understanding?

Author Response

Thank you very much for your valuable and significant comments. Based on the comments received, we had discussed several times and understood the reviewers’ multiple concerns. Our paper changed into a better version while considering all the comments which the reviewer pointed out. Please see our response in the table below and find newly changed parts of red letters in our revised manuscript. Again thanks.

This article deals with effects of short-term and long-term hearing changes on music exposure for young adults. Using strategy for systematic search, authors selected 16 articles and analyzed the context of these articles. They concluded that personal listening devices (PLDs) users seem to have a few hearing changes at 4kHz after use, and repetitive and cumulative use will connect to permanent hearing changes or loss. These main research findings will be important for understanding effects of music exposure in people. The study design, procedure, data analysis, and conclusions appropriate. Minor revisions may be needed

Point 1: Minor revision suggestions: Some figures (from figure 2 to figure 5) is unclear and hard to understand. Could you organize the data in figures and provide additional explanation for easy understanding?

Response 1: We appreciate your valuable and affirmative comments. We agreed with your suggestion and thus added some descriptions related data in the figures 2 to 5, while providing additional explanations for better understanding of the current results. Specifically, the effect sizes which were not stated in the results, such as PTA thresholds at 6 and 8 kHz, and their relationship were provided in the revised manuscript, using Track Changes of the red colours.

Line 173 to 175

  • Interestingly, the thresholds at 6 and 8 kHz showed different aspects of effect size. The effect sizes of 6 and 8 kHz thresholds appeared relatively smaller; 6 kHz: -0.059, CI: -0.438 to 0.320, p: 0.760 and 8 kHz: -0.066. CI: -0.436 to 0.303, p: 0.725. Moreover, the 6 and 8 kHz thresholds…

Line 182 to 184

  • Although amplitude at 6 kHz showed a similar effect size with 4 kHz (0.106, CI: -0.065 to 0.278, p: 0.224), the amplitude at 2 kHz was relatively less affected (0.029, CI: -0.154 to 0.212, p: 0.755).

Reviewer 2 Report

In this paper the authors evaluated whether music exposure temporally or permanently affects hearing sensitivity in young adults. Text is well written. Figures and tables are explicative and adequate. The authors should better elaborate in the introduction the risks of leisure noise exposure among children and teenagers - see Ralli M, Greco A, de Vincentiis M. Hearing Loss Following Unsafe Listening Practices in Children, Teenagers and Young Adults: An Underestimated Public Health Threat, Int J High Risk Behav Addict. 2018 ; 7(3):e65873. doi: 10.5812/ijhrba.65873. Last, english language should be checked for minor errors.

Author Response

Thank you very much for your valuable and significant comments. Based on the comments received, we had discussed several times and understood the reviewers’ multiple concerns. Our paper changed into a better version while considering all the comments which the reviewer pointed out. Please see our response in the table below and find newly changed parts of red letters in our revised manuscript. Again thanks.

Response to Reviewer 2 Comments

In this paper the authors evaluated whether music exposure temporally or permanently affects hearing sensitivity in young adults. Text is well written. Figures and tables are explicative and adequate.

Point 1: The authors should better elaborate in the introduction the risks of leisure noise exposure among children and teenagers - see Ralli M, Greco A, de Vincentiis M. Hearing Loss Following Unsafe Listening Practices in Children, Teenagers and Young Adults: An Underestimated Public Health Threat, Int J High Risk Behav Addict. 2018 ; 7(3):e65873. doi: 10.5812/ijhrba.65873.

Last, english language should be checked for minor errors.

Response 1:  Thanks for your positive comments. The authors had read the Ralli et al.’s paper which you suggested and included it in the introduction related the music exposure to children and teenagers. Thus, the revised manuscript has a state of problems with specific values such as the prevalence of hearing loss in young adults.

Also, we had consulted our revised manuscript with a native English speaker having long career in professional editing and improved it in terms of English grammar and composition. Attached are files for its invoice. Please find many red letters in the revised manuscript. Thanks again.

Line 47 to 49

In addition, the prevalence of hearing loss with music exposure in young adults ranges from 9.3% to 14% depending on the study or nation [7]. In addition…

[7]: Ralli, M., Greco, A., & de Vincentiis, M. (2018). Hearing Loss Following Unsafe Listening Practices in Children, Teenagers and Young Adults: An Underestimated Public Health Threat. International Journal of High Risk Behaviors and Addiction7(3).

Line 63

  • User -> young adults’

Reviewer 3 Report

I think this is an interesting and solid literature review on an important topic. The methodology is rigorous and PRISMA and PICOS guidelines are followed carefully.

English is generally fine, but please check again as I have spotted several typos. 

In the Introduction, I would like the authors to put their aims in context, in particular by highlighting the policy relevance of their review. In the WHO Environmental Noise Guidelines for the European Region (2018), the "leisure noise" is a category of exposure introduced to cover personal devices for music and similar, as those described by the authors, and limits are discussed with (conditional) recommendations. I would like to see this systematic review discussed in this context.

Also, I would recommend a trained acoustician to check carefully how noise exposure units are reported. I have noticed significant inconsistencies in the notation (in Table 3 in particular, but also throughout the text). E.g., Laeq, LAeq, dBSPL (?), dB(A) dBA, dBC, dB(C), etc. Please revise and make sure frequency weightings, subscripts for temporal averaging, italics, and everything else is reported consistently as per the standard of the sector.

Overall, very interesting reading. 

Author Response

Thank you very much for your valuable and significant comments. Based on the comments received, we had discussed several times and understood the reviewers’ multiple concerns. Our paper changed into a better version while considering all the comments which the reviewer pointed out. Please see our response in the table below and find newly changed parts of red letters in our revised manuscript. Again thanks.

Response to Reviewer 3 Comments

I think this is an interesting and solid literature review on an important topic. The methodology is rigorous and PRISMA and PICOS guidelines are followed carefully.

Point 1: English is generally fine, but please check again as I have spotted several typos. 

Response 1: Thanks for your wonderful comments. As your suggestion, we carefully checked the entire manuscript in terms of English grammar and composition, and we corrected several typos from your points. In addition, a native English speaker with long careers in professional proof editing reviewed the revised manuscript before the submission. Attached are files for its invoice. Please see our revised manuscript with correction of the red colours. Thanks again.

Point 2: In the Introduction, I would like the authors to put their aims in context, in particular by highlighting the policy relevance of their review. In the WHO Environmental Noise Guidelines for the European Region (2018), the "leisure noise" is a category of exposure introduced to cover personal devices for music and similar, as those described by the authors, and limits are discussed with (conditional) recommendations. I would like to see this systematic review discussed in this context.

Response 2: Thanks for your valuable comments and effort. The suggested literature (WHO guideline) was carefully reviewed from all authors, to highlight the purpose and direction in the introduction of the current review paper. We added some sentences in revised manuscript while using a function of Track Change and red colour.

Line 222 to 228

Similar to the present study, the World Health Organization (WHO; 2018) reported guidelines for leisure noise with regulation of exposure intensity and time, health outcomes, and interventions [36]. In its assessment of the evidence on hearing loss, WHO determined that (1) the effect of regular music exposure cannot be quantified, (2) the effect of specific threshold analysis is limited, and (3) analysis of the pooled effect size is limited due to lack of data from previous studies. Although the present study cannot resolve all the limitations in the guidelines, it can present insight into behavioral and objective hearing outcomes at a specific frequency based on a meta-analysis.

Line 247 to 253

  • Although limited to direct comparison to WHO’s regulation [36] due to heterogeneity such as difference in units and time, only two studies were comparable [22, 27]. Based on LAeq,1h, Keppler and Bockstael (2010) reported 92.41 dB LAeq,1h at one hour [22] and Keppler et al. (2014) reported 86% of subjects exposed at greater than 92.25 dB LAeq,1h at one hour [27]. Considering the hazardous exposure intensity and time in the regulation (95 dB LAeq, one hour), these results indicating no significant change in PTA or DPOAE were consistent with the regulation.
  •  

Line 287 to 293

  • Similar to investigation of the short-term effect, only three studies were compared against the WHO regulation [36]. In terms of PTA, based on LAeq, Kumar and Deepashree (2016) reported 87.97 dB LAeq at two hours per day [31] and Hussain et al. (2018) reported 83 dB LAeq at 1.99 hours per day [33]. This is in comparison to the regulation (74 dB LAeq, two hours per day). These results exceed the regulation and are consistent with the significant poor PTA results. Furthermore, Torre and Reed (2019) reported 73.0 dBA LAeq at one hour per day, which did not exceed the regulation, with no significant DPOAE amplitude [35].

Point 3: Also, I would recommend a trained acoustician to check carefully how noise exposure units are reported. I have noticed significant inconsistencies in the notation (in Table 3 in particular, but also throughout the text). E.g., LAeq, LAeq, dBSPL (?), dB(A) dBA, dBC, dB(C), etc. Please revise and make sure frequency weightings, subscripts for temporal averaging, italics, and everything else is reported consistently as per the standard of the sector.

 Overall, very interesting reading. 

Response 3: Thanks for your suggestion. We agreed with your points in that the units and notations were inconsistent among the studies. However, the intensity data of the pooled studies which we selected in this systematic review could not be synthesized because of heterogeneity of outcome measurement (specifically measurement unit) and measurement methods vary depending on the study. More specifically, these intensity results in various studies (e.g., dB SPL, dBA, dBC, LAeq, etc.) could not be easily integrated due to the different weights and temporal averaging according to frequencies when it was measured. Although it might be very careful to integrate differently measured intensity units for each study to use meta-analysis, we added our limitation of study in the Discussion section. Thank you again for your great comments and consideration on our manuscript.

Line 295 to 302

4.3. Limitation of the study

  • The present study has limitations in terms of inconsistency in units. The intensity data of the pooled studies which we selected in this systematic review could not be synthesized because the heterogeneity of outcome measurement (specifically the measurement unit) and the measurement methods vary depending on the study. More specifically, the intensity results in various studies (e.g., dB SPL, dBA, dBC, LAeq, etc.) cannot be easily integrated due to the different weights and temporal averaging according to frequency when first measured. However, it might be interesting to integrate the differently measured intensity units for each study to use meta-analysis, so further investigation should be considered.